# Urban Sprawl Patterns, Drivers, and Impacts: The Case of Mogadishu, Somalia Using Geo-Spatial and SEM Analyses

**Marwan Omar Hassan** [1] , **Gabriel Hoh Teck Ling** [1,*] , **Noradila Rusli** [2] , **Safizahanin Mokhtar** [1] , **Walton Wider** [3,*] and **Pau Chung Leng** [1]

1 Faculty of Built Environment and Surveying, Universiti Teknologi Malaysia, Johor Bahru 81300, Malaysia
2 Centre for Innovative Planning and Development (CIPD), Department of Urban and Regional Planning, Faculty of Built Environment and Surveying, Universiti Teknologi Malaysia, Johor Bahru 81300, Malaysia
3 Faculty of Business and Communications, INTI International University Persiaran Perdana BBN, Putra Nilai, Nilai 71800, Malaysia
* Correspondence: gabriel.ling@utm.my (G.H.T.L.); walton.wider@newinti.edu.my (W.W.)

**Abstract:** There is a lack of research on urban sprawl in developing countries, particularly in Sub-Saharan Africa, undergoing significant demographic change. There is an urgent need to conduct more studies on African cities and investigate spatial variations in urban sprawl to fill a knowledge gap in Sub-Saharan Countries (SSC). There have been no studies of urban sprawl in the Somali capital of Mogadishu, a fragile metropolis struggling with the legacy of decades of civil war. This study has two main objectives: (i) to examine sprawl patterns in Mogadishu, Somalia; and (ii) to identify the drivers and impacts of urban sprawl in Mogadishu, Somalia. The study used spatiotemporal imagery from 2006, 2013, and 2021 to identify sprawl patterns. A quantitative method in the form of a cross-sectional survey with 265 participants was then used to identify the drivers and impacts of sprawl, which was then analysed using the structural equation model (SEM). The spatiotemporal analysis results showed sprawl patterns in nine districts and three settlements, mainly scattered and leapfrog patterns. The SEM discovered five significant drivers: low price of land and dwelling (LP), development of transportation infrastructure (DTI), rising income, security reasons, and low commute cost (LCC), in addition to eight significant impacts: less social interaction (LSI), agriculture land and natural habitat loss (AGL NHL), unsafe environment (USE), insufficient health and educational services (IHF IEF), high public services cost (HPSC), insufficient public transport (IPT), less physical activity (LPA), pollution (POL) and mental health issues (MH). Undoubtedly, the impacts found in the study proved that urban sprawl negatively impacted the residents and environment of Mogadishu, which will continue as the security situation in the city improves and more residents are attracted.

**Keywords:** urban sprawl; urban sprawl drivers; urban sprawl impacts; spatiotemporal analysis; Sub-Saharan Africa

## 1. Introduction

Urbanisation is a crucial driver of city growth, but if not handled effectively, it jeopardises urban sustainability [1]. As a result of urban development and urbanisation, the city's space and the places it uses as urban space are expanding, dispersing, and altering the usage of formerly rural and natural regions and landscapes [2]. Urban sprawl is one of the issues in today's cities [3]. The term "Urban Sprawl" refers to excessive expansion, which distinguishes it from other types of urban growth [3]. While Gorden describes the phenomena as having a horrible growth pattern expanding further from the metropolis, such as leapfrogging, ribbon, strip development, and being scattered [4]. It causes various issues such as inadequate connection, diminished or costly public services [5,6], higher energy use and transportation [7], air pollution and traffic congestion [2], and permanent environmental degradation [8]. It also impacts the quality of life [9] and considerably influ-

ences the unit cost of local public services, resulting in an inefficient urban development model [10].

Numerous theoretical and practical studies have been conducted on urban sprawl worldwide, and planners and lawmakers are increasingly concerned about urban sprawl [11]. From the 1960s to the present, an enormous body of literature has accumulated on the drivers and effects of sprawl. The vast majority of that research was conducted in developed countries [12]. However, there is a general lack of research focusing on cities in developing countries. More research is needed to understand the reasons for sprawl in these places [13]. Developing nations are losing one to two million hectares of farmland per year from their prime agricultural land to meet the growing need for housing, infrastructure, and leisure [1]. Many other academics have stated that research on urban sprawl in developing nations, particularly in Sub-Saharan Africa, is insufficient [14]. There is a major demographic shift [15], and fast urbanisation, which threatens to jeopardise the achievement of the Sustainable Development Goals (SDGs), particularly Goal 11, which calls for the creation of liveable and sustainable cities and communities [14]. Moreover, the drivers and impacts of sprawl differ between and within Sub-Saharan African nations [16]. As a result, it is imperative to conduct more studies in Sub-Saharan African cities to fill the aforesaid knowledge gap [14]. Somalia is a member of the SSC, with Mogadishu as its capital. Mogadishu is a fragile metropolis struggling with the legacy of decades of civil conflict in a political setting marked by shaky elite bargaining and an imperfect constitutional transition [17], where no urban sprawl research is carried out in Somalia in general or Mogadishu.

This paper will fill the empirical and knowledge gap by attempting to answer three research questions. First, where are the urban sprawl areas and what are the sprawl patterns in Mogadishu? Second, what are the drivers of urban sprawl in Mogadishu? Third: what are the impacts of urban sprawl in Mogadishu? This research will increase the knowledge of urban sprawl in Sub-Saharan nations and highlight a problem not mentioned in Somalia's literature. This will enable Somalia's government, planning agencies, and international organisations to build effective urban sprawl mitigation policies and guidelines.

## 2. Literature Review

### 2.1. Urban Sprawl

Urban planners were the first to perform research on urban sprawl, which is a multi-disciplinary issue. As a physical phenomenon, urban sprawl covers various disciplines: environmental studies, urban planning, geography, sociology, economics, and even policy science. Numerous definitions attempting to capture the urban sprawl phenomenon's complexity and interdisciplinary character have been developed due to considering many different factors. Often, these meanings conflict, resulting in misunderstanding [12]. The idea of urban sprawl encompasses various dimensions, illustrating how urban built-up areas accumulate throughout exurban environments [18]. The higher the degree of urban sprawl, the more territory is built up, the more dispersed the structures, and the more visible it is in the landscape [12]. Peiser was among the first to argue that unchecked growth causes urban sprawl [19]. According to Peiser [19], urban sprawl has become the dominating pattern of growth. Among the evidence are the poor management of land use change and the boundary for environmentally sensitive areas. As Peiser pointed out, out-of-control development is linked to early conversion of rural and agricultural land use, inadequate urban land use planning about surrounding benefits, and urban land use that was not effectively developed with public services and infrastructure [18].

### 2.2. Urban Sprawl Spatial Identification

Knowledge of the existing patterns of urban land use, as well as the trends in urban sprawl intensity and direction, is essential [20]. It is difficult to estimate urban sprawl using standard surveying and mapping procedures, which are costly and time-consuming, particularly in developing nations [21]. However, technological innovations such as remote sensing (RS) and geographic information systems (GIS) are frequently utilised to monitor

and track urban sprawl [22], making monitoring and detecting land use and land cover (LULC) easier for investigating lengthy periods and covering enormous geographic regions [23]. Additionally, the built-up area is an adequate measure of urban sprawl since the built-area change over time exposes the type, breadth, and evolution of urban sprawl [22]. Urban sprawl is found on the city's periphery [24].

Generally speaking, it is very simple to see the spatial aspects of urban sprawl [25]. It has a horrible growth pattern expanding further from the metropolis, such as low-density, leapfrogging, ribbon, strip development, and being scattered [4]. Leapfrog development refers to a form of urbanisation in which newly built regions are located in isolation from one another and the established limits of larger cities [26]. It happens outside the urban borders of the metropolis [27]. It occurs because developers prefer to construct on cheaper land further from the city centre rather than the city centre's expensive land [26]. The term "ribbon development" describes the growth of cities along transportation corridors. In many instances, ribbon development is located on the periphery of residential centres, although in some other cases, ribbons look isolated and form the basis of the built environment [28].

### 2.3. Urban Sprawl Drivers

Every city, state, and continent has its unique drivers of urban sprawl, which varies greatly from place to place. As a result, urban sprawl is subject to a wide range of drivers [18]. As with any other kind of urbanisation, sprawl is primarily driven by population expansion [29–32]. Most experts believe that government policies, the expansion of the highway system, the widespread use of vehicles, economic success, and the democratisation of society have all contributed to urban sprawl development [13]. Urban sprawl drivers can be categorised as a combination of demographic, socioeconomic and political drivers. Demographic drivers have been identified as one of the primary causes of urban sprawl [18]. Among demographic drivers increasing population and migration have been pinpointed as crucial factors contributing to urban sprawl [33,34]. Additionally, low commuting costs [35], rising income [18,29], the cheaper value of agricultural land [34,35], and employment possibilities and the availability of affordable dwellings are among the socioeconomic drivers. Nevertheless, institutional drivers have a critical effect on urban sprawl [36]. There is a variety of institutional drivers such as poor or lack of master planning [35,36], lack of control of illegal dwellings [37], land speculation [26] and even decentralised governance [38] and democratisation of society [13].

### 2.4. Urban Sprawl Impacts

Sprawl has a variety of environmental, socioeconomic, and economic benefits and drawbacks for both urban and rural populations. However, a sprawling metropolis causes environmental, social, and economic concerns [5]. It also poses a public health risk and hurts the national and local economies [5], in addition to impacting people's quality of life [9]. Feng and Gauthier [39] showed that urban sprawl has a significant negative impact on the environment, including the acceleration of global climate change [39]. Additionally, among the environmental impacts is the significant transformation of agricultural lands into urban areas and the loss of natural resources and water bodies [40,41]. Urban sprawl has significant and detrimental effects on socioeconomics such as higher public services and household costs [42] and the absence of essential services [43], in addition to less social interaction [5] and social segregation [44]. Urban sprawl also impacts health, causing an increase in respiratory diseases [5], risk factors for chronic illnesses [29], and mental health [7] and contributing to diabetes, obesity, and high blood pressure [29]. Urban sprawl can also cause pollution, jeopardising people's health [1]. Nevertheless, urban sprawl also impacts the quality of life of those who live in sprawled areas as they have a higher risk of traffic deaths, poor walking conditions due to a lack of sidewalks, and slower public transportation since they travel longer distances [45], and have unclean water and bad sanitation [9].

## 3. Study Area, Data, and Methods

### 3.1. Mogadishu, Somalia

The capital of Somalia, Mogadishu, is the country's largest city and can be found in the Banadir coastline area (see Figure 1). It is Somalia's commercial harbour, and it has 17 administrative districts [46]. It is located between 2°2′48.9624″ N and 45°19′5.3796″ E. Mogadishu is a vulnerable metropolis struggling with the aftermath of decades of civil conflict in a political environment marked by shaky elite agreements and an imperfect constitutional transition [17]. It has been the main battleground for clan-based warlords, terrorist organisations, and the Somali government [46]. Security is shifting as Al-Shabaab (AS) terrorists attack government buildings and foreign organisations, not civilians, in the city [47]. The city's population has grown since the transition due to greater security and the private sector's revival. Markets are bustling, new companies are opening, tourism is booming, and locals can walk securely [47].

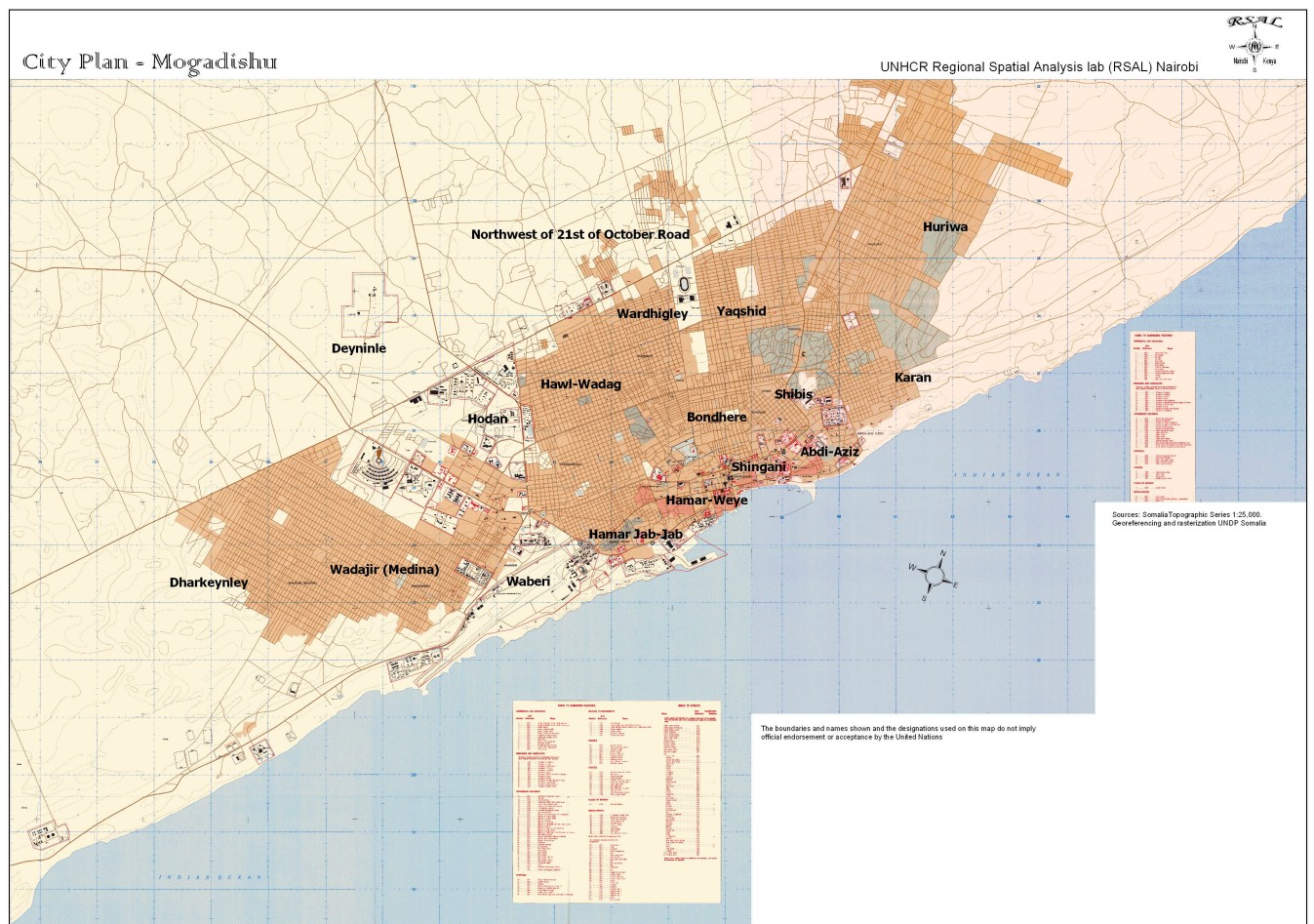

**Figure 1.** Map of Mogadishu (UNHCR, 2014).

History

It is thought that Mogadishu's history dates back to at least the 10th century. Ethnic Somalis live in neighbouring countries such as Kenya, Ethiopia, and Djibouti [47]. Xamar Weyne was first colonised in the early 10th century by migrants from Yemen, Oman, and Persia. Toward the end of the 1300s, Mogadishu became the most important east African commerce city for Arabs [48]. Mogadishu continued growing economically and demographically for the following centuries. Mogadishu was proclaimed Italian-controlled in 1889 as the Italians began exerting their authority in the city [48]. Somalia was separated during colonial times into northern British Somaliland and southern Italian Somaliland [47].

On 26 June 1960, the United Kingdom gave independence to the northern territories. Italian-Somali independence was granted four days later. The Somali Republic was established on 1 July 1960, when the inhabitants of the former British and Italian colonies united [47].

Furthermore, according to Grunewald [49], during the Italian colonial period, there was only a small amount of urbanisation. Under the leadership of Governor Guido Corni, the first urban planning practice was held in 1937. This is the only city planning project in the Somali capital that has ever been completed. The city expanded outside of its historic core during the years 1969–1991 under the leadership of the Siyad Barre regime. Meanwhile, the deep-sea port developed into a bustling commercial centre, luring an increasing number of unskilled workers, many of whom were impoverished nomads and agro-pastoralists fleeing drought and other economic hardships in the countryside [49].

Moreover, Mogadishu had 90,000 residents at independence. By the 1980s, soon before Siad Barre's fall, it had swelled to one million, with unplanned informal settlements mushrooming and residents living in cramped, unsanitary circumstances without essential utilities. The centre area doubled from 1970 to 1984, and at that time, the city had 13 administrative districts. Subsequently, after the fall of the Siad Barre government in 1991–1992, Mogadishu's centre was mostly damaged by intra-clan violence. During this time, the city's services were almost completely wiped out, and it was split into a network of warlord-controlled fiefdoms, which lasted 15 years [17]. This continued until 2006, when the Islamic Courts Union (ICU), an alliance of business owners and local Islamic leaders championing the primacy of Sharia law over clan rules, seized control of the majority of the south and central areas and reopened the port of Mogadishu [49].

Later, after the ICU seized control of Mogadishu in 2006, only to be ousted the following year by Ethiopian forces, Mogadishu was once again the site of conflict. In 2007 and 2008, the city sustained more damage as a result of conflicts between the transitional government and Al-Shabaab (AS), a militia allied to Al Qaeda, until the latter withdrew from the majority of Mogadishu in 2008 under pressure from the African Union Mission in Somalia [17]. Urban fighting between multinational forces resumed in Mogadishu. Mogadishu's residents went through a terrible period. As urban warfare is predicated on massive artillery shelling and street-by-street combat among foes, the wreckage created by these clashes is obvious in many regions [49]. Finally, despite the possibility of terrorist attacks, Mogadishu appears to be peaceful and quiet, as seen by the bustling trade, the return of Somalis from the diaspora to Somalia, and the presence of foreign players such as the United Nations, which had previously been absent from the city [46]. An interim national authority administered Somalia from 2004 until 2012 when it formally became the Federal Government of the Federal Republic of Somalia (FRS) [47].

*3.2. Methods*

Urban Sprawl Identification

The urban expansion must be recorded and monitored across several temporal dimensions to understand Mogadishu's sprawl. This study utilised ArcGIS pro to detect urban sprawl in Mogadishu and its patterns in the years 2006, 2013, and 2021. Satellite photos were obtained from Google Earth at a maximum resolution of 8192 × 5134 pixels. The training samples used for each class were 40 samples. Furthermore, the analysis was carried out utilising an image classification method and a maximum likelihood algorithm. Two categories were used to classify the research area: developed and underdeveloped (Table 1).

**Table 1.** Land use classes and definitions used in the classification.

| | Classes | Definitions |
|---|---|---|
| 1 | Developed | The area consists of buildings and developed roads. |
| 2 | Undeveloped | The area consists of bare lands, vegetation, and water bodies. |

*3.3. Urban Sprawl Drivers and Impacts*

3.3.1. Data Collection

This study investigated urban sprawl drivers and impacts by employing a quantitative method in the form of a cross-sectional survey. The research can only address a few of the many factors and impacts discussed in the literature. Thus, after carefully studying Mogadishu's history, the study hypothesised eight drivers and thirteen impacts of urban sprawl in Mogadishu (Table 2). The questionnaire was created using Google Forms. The questionnaire questions consist of three main sections, and each portion is broken down into a series of questions. The first section covers demographics and socioeconomics, the second covers urban sprawl drivers, and the third covers its impact. Multiple-choice questions will be utilised to respond to the first segment, while the Likert scale will be used for the second and third.

**Table 2.** Hypothesised urban sprawl drivers and impacts of Mogadishu, Somalia.

| Number | Urban Sprawl Drivers | Urban Sprawl Impacts |
| --- | --- | --- |
| 1 | Low prices of land or dwelling | Social segregation |
| 2 | Rising income | Less social interaction |
| 3 | Security reasons | Higher household cost |
| 4 | Migration from inner-city | Higher public services cost |
| 5 | Family nuclearisation | Loss of natural habitats |
| 6 | Low commuting cost | Agriculture land loss |
| 7 | Development of transportation infrastructure | Insufficient public transport |
| 8 | | Insufficient educational facilities |
| 9 | | Insufficient health facilities |
| 10 | | Unsafe environment |
| 11 | | Less physical activity |
| 12 | | Mental health issues |
| 13 | | Pollution |

3.3.2. Sampling Size and Methods

The sampling method used was snowball sampling, where the rationale for choosing was due to the difficulty of covering the 9 districts and 3 settlements where sprawl was found, so participants from sprawled areas were assigned to distribute the questioner. In addition, the households of Mogadishu's sprawled area have been chosen as the study's target demographic. The reason for selecting this group of respondents was because the inhabitants of the city-sprawled area are the people facing the investigated phenomena. Furthermore, in every quantitative study with a cross-sectional survey design, numerous strategies have been utilised to determine the suitable sample size. Because the size of the population living in urban sprawl areas has yet to be discovered, it is hard to determine the exact sampling size; however, the Cochran formula is the most widely used [50]; thus, the study used the Cochran formula with $p = 0.05$, 95% confidence level, and ±5% precision, and the desired sample size was determined at 385 participants. A total of 265 people participated in the survey, which spanned a period of 22 days, from 26 November 2022 to 18 December 2022.

3.3.3. Data Analysis

This study used structural equation modelling, a frequently used statistical modelling approach in the social sciences. It may be thought of as a hybrid of regression and factor analysis [51]. The purpose of SEM is to examine a set of relationships between one or more exogenous variables (independent variables) and one or more endogenous variables (dependent variables) [51]. This study first performed exploratory factor analysis (EFA) to test the measuring instrument and minimise the number of factors using SPSS 26. Then, the Amos software was utilised to perform confirmatory factor analysis (CFA) and SEM.

## 4. Results and Findings

### 4.1. Urban Sprawl Pattern

As presented, the city in 2006 was densely concentrated on the southern side (Figure 2). There was a sprawling development pattern in districts such as Hilwaa, Wadajir, and Dharkeenley, especially in Hilwaa, where the built-up areas were around the Balcad main road, which separates Hilwaa and what is now known as the Darussalam settlement. The Balcad main road development pattern can be classified as having a sprawling ribbon pattern going to the northeast. However, it can also be classified as a scattered development as the development pattern is not only attached to the Balcad road. The remaining districts also sawed a scattering pattern in the built-up area periphery.

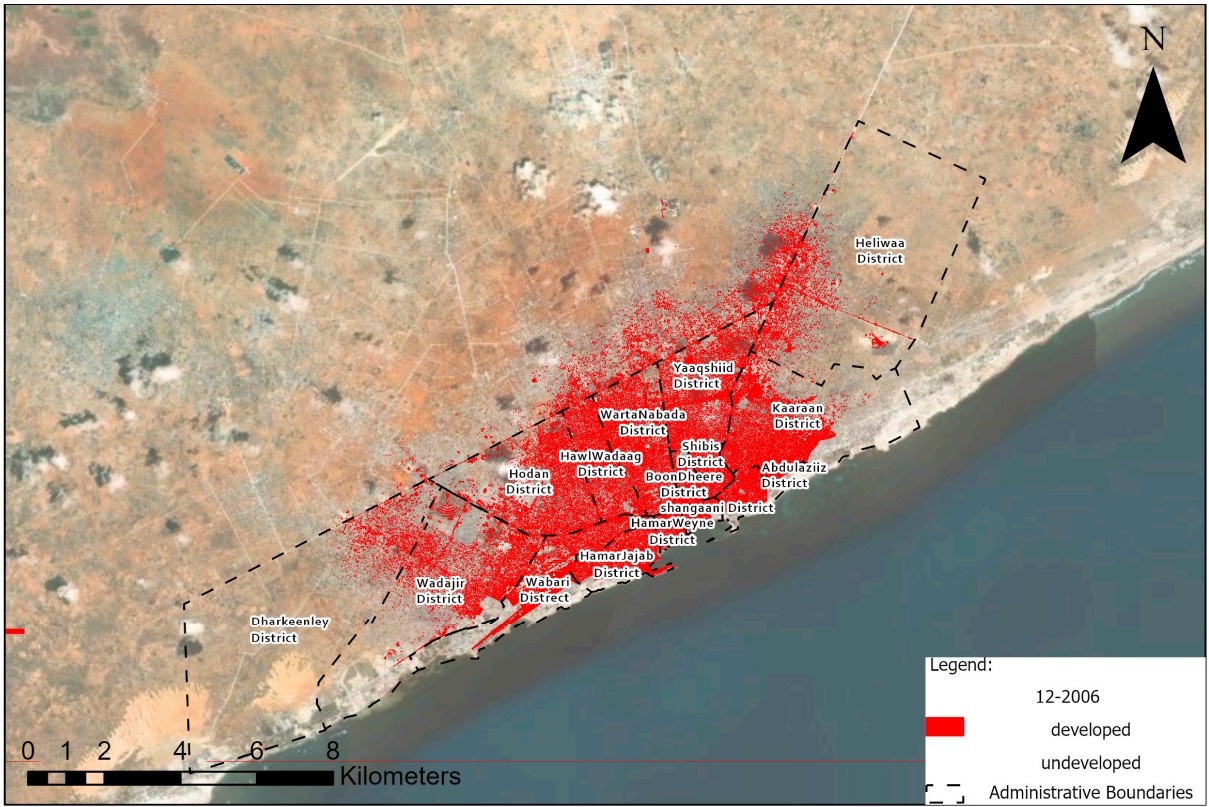

**Figure 2.** The built-up area of Mogadishu city in the year 2006.

Moreover, the 2013 built-up area (Figure 3) shows an increase in the built-up area, especially in Wadajir and Hodan and what is now known as Garasbaaleey. The development pattern continued scattered along the periphery districts and settlements; however, the development in Garasbaaley is presented in a leapfrog combined with scattered development. Additionally, the 2021 built-up area pattern shows that the Garasbaaley built area changed as previously developed areas disappeared. However, the area development shows a massive expansion with a leapfrog and scattered pattern. Additionally, the majority of growth was towards the northwest and a slight increase in the east.

Additionally, the 2021 built-up area pattern (Figure 4) shows that some districts showed a considerable built-up area expansion, such as Kaaran, Wadajir, and Dherkeenly. Furthermore, the rapid expansion of districts resulted in the creation of multiple settlements such as Daarusalaam, Garisbaaley, and Gudadley and what is also known as the Afgooye corridor. The Garasbaaley built area changed as previously developed areas disappeared. However, the area development shows a massive expansion with a leapfrog and scattered pattern. The majority of the expansion growth is towards the northwest, and a slight increase in the east.

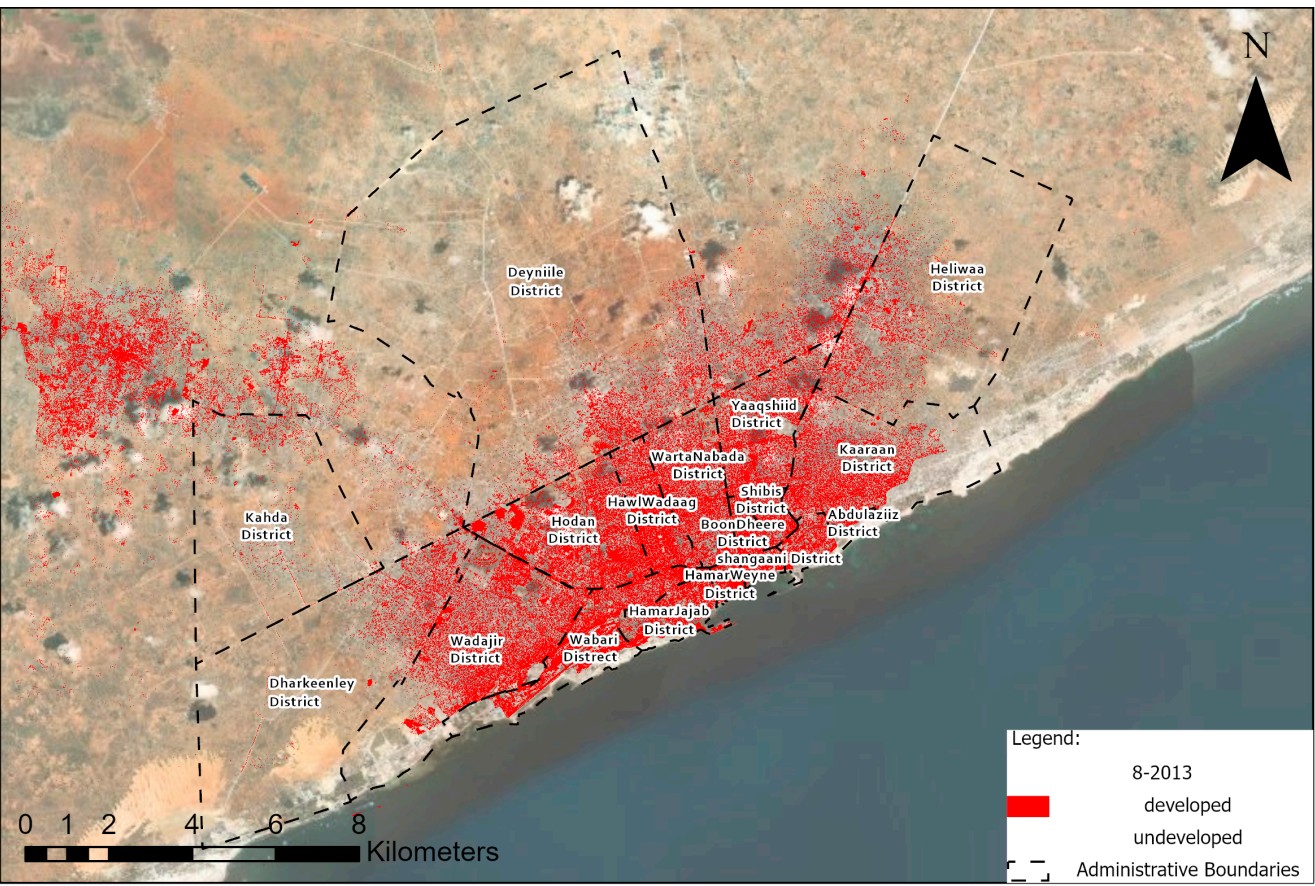

**Figure 3.** The built-up area of Mogadishu city in the year 2013.

The overlaying map of the three produced classifications in Figure 5 showed that the expansion in Mogadishu is seen to have different sprawling patterns, such as leapfrog, scattered, and ribbon. The main urban sprawl patterns noticed are leapfrog and scattered, which were located in the northeast and the north. In summary, urban sprawl is found in all the districts and settlements except HamarJajab, Hamrwayne, Shibis, Abdiaziz, Shangaani, Wartanabada, Waaberi, Howlwadaag and Boondheere. As the remaining show different degrees of sprawl, they thus are ideal for studying urban sprawl drivers and impacts in Mogadishu city.

Accuracy Assessment

To evaluate the accuracy of the image's classification, the classified maps of 2006, 2013, and 2021 are compared to Google Earth imagery. Each was given a total of 500 accuracy points using the stratified random approach in Arc GIS pro-2.8. Then, these points were placed over the categorised picture, and a confusion matrix was generated. Subsequently, the user accuracy, producer accuracy, overall accuracy, and Kappa coefficient are computed. The findings of Kappa values greater than 0.8 indicate that the categorisation falls into the very good category (Table 3).

**Table 3.** Accuracy assessment of classified images.

| | Category | 12-2006 | 8-2013 | 12-2021 |
|---|---|---|---|---|
| Producer Accuracy | Developed | 76.9% | 89.7% | 94.8% |
| | Un-Developed | 98.8% | 98.6% | 97.8% |
| User Accuracy | Developed | 88.8% | 85.3% | 89.1% |
| | Un-Developed | 97.3% | 99.1% | 99.0% |
| Overall Accuracy | | 96.0% | 98.0% | 97.4% |
| Kappa coefficient | | 0.806 | 0.864 | 0.903 |

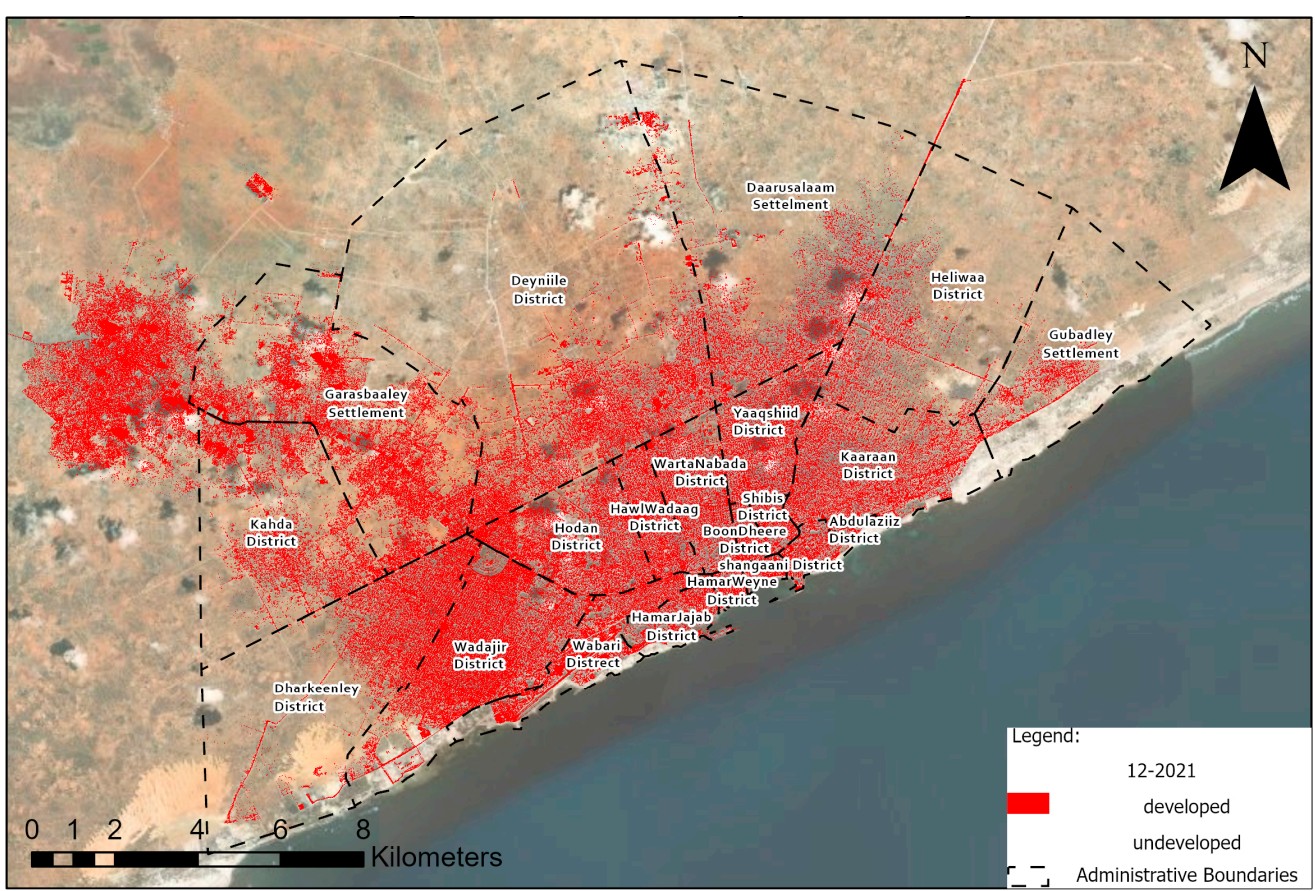

**Figure 4.** The built-up area of Mogadishu city in the year 2021.

*4.2. Descriptive Statistics*

4.2.1. Participants' Profiles

The study sample size was 265 participants from the locations of urban sprawl. Overall, 67.9% of participants were male, while 32.1% were female (Table 4). Most participants were 21–30 years old. Additionally, 39% of participants reported a monthly household income between USD 301 and 600, 26.8% reported less than USD 300, and 19.2% reported USD 601–900. Beyond that, 7.2% reported a USD 901–1200 income, while 7.5% reported a higher income. Additionally, the survey included participants from eight districts and two settlements, mainly from Hodan and Wadajir. Overall, 25.7% of participants resided in their present residence for fewer than five years, followed by 25.3% for 5–10 years, while 49% had lived there for over ten years. The majority of the participants were living in ground-floor villas.

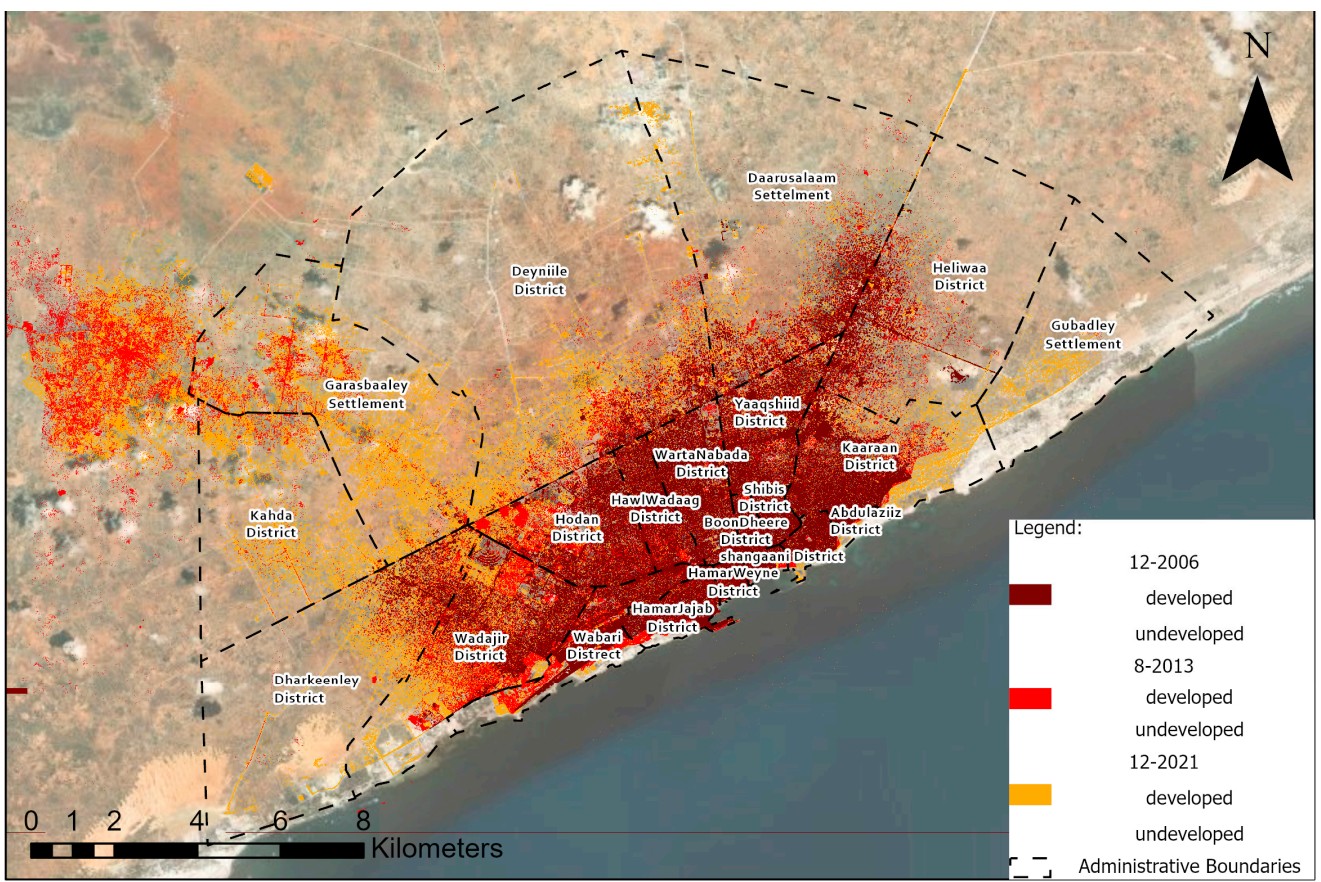

**Figure 5.** The built-up area of Mogadishu city in the years 2006, 2013 and 2021.

**Table 4.** Participants' descriptives.

|  | Category | Number of Participants | Percentage |
|---|---|---|---|
| Gender | Female | 85 | 32.1% |
|  | Male | 180 | 67.9% |
| Age | 15–20 years old | 51 | 9.1% |
|  | 21–30 years old | 186 | 70.2% |
|  | 31–40 years old | 24 | 19.2% |
|  | +40 years old | 4 | 1.5% |
| Income | USD 100 to 300 | 71 | 26.8% |
|  | USD 301 to 600 | 104 | 39.2% |
|  | USD 601 to 900 | 51 | 19.2% |
|  | USD 900 to 1200 | 19 | 7.2% |
|  | More than USD 1200 | 20 | 7.5% |

**Table 4.** *Cont.*

| | Category | Number of Participants | Percentage |
|---|---|---|---|
| | Dyniile district | 24 | 9.1% |
| | Wadajir district | 63 | 23.8% |
| | Hilwaa district | 7 | 2.6% |
| | Yaqshiid district | 49 | 18.5.% |
| Living place | Hodan district | 70 | 26.4% |
| | Karaan district | 13 | 4.9% |
| | Kahda district | 2 | 0.8% |
| | Dharkeenley district | 23 | 8.7% |
| | Garasbaaley settlement | 8 | 3.0% |
| | Daarusalaam settlement | 6 | 2.3% |
| | Less than 5 | 68 | 25.7% |
| Living duration | 5–10 years | 67 | 25.3% |
| | More than 10 years | 130 | 49.0% |
| | Metal zinc homes | 50 | 18.9% |
| Housing type | Ground floor villa | 160 | 60.4% |
| | Multi-story villa | 22 | 8.3% |
| | Apartment | 33 | 12.5% |

### 4.2.2. Urban Sprawl Drivers

The findings of the study show that drivers such as low prices of land, migration from the inner city, and the development of transportation infrastructure showed high agree and strongly agree percentages in the total of 41.8%, 39.3%, and 38.3%, respectively (Table 5). The remaining drivers showed less than 30% total agree and a strongly agree percentages. On the other hand, drivers such as family nuclearisation, low commuting cost, and security reasons showed high disagreement and strong disagreement, with totals at 56.5%, 50%, and 44.8%, respectively.

**Table 5.** Urban sprawl drivers' descriptive results.

| Driver | Questions (Items) | Percentages | | | | | Mean | S.Deviation |
|---|---|---|---|---|---|---|---|---|
| | | Strongly Disagree | Disagree | Neutral | Agree | Strongly Agree | | |
| | Q1- I moved to my current location because I couldn't afford to live in the inner city. | 11.7 | 28.7 | 21.9 | 22.6 | 15.1 | 3.01 | 1.261 |
| Low prices of land or dwelling | Q2- I moved to my current location because the land or dwelling price is far less than in the inner city. | 11.7 | 30.9 | 17.4 | 23.0 | 17.0 | 3.03 | 1.301 |
| | Q3- I moved to my current location because I wanted bigger land at a lower price. | 11.3 | 23.0 | 17.7 | 27.9 | 20.0 | 3.22 | 1.311 |
| | Q1. My income rose before I moved to my current location. | 20.4 | 30.6 | 24.5 | 16.6 | 7.9 | 2.61 | 1.208 |
| Rising income | Q2- I moved to my current location because I wanted a better standard of living now that I can afford it. | 15.5 | 26.0 | 28.3 | 18.5 | 11.7 | 2.85 | 1.231 |
| | Q3- I moved to my current place because I could afford to live in a bigger dwelling after my income rose. | 17.0 | 26.8 | 22.6 | 21.5 | 12.1 | 2.85 | 1.276 |

**Table 5.** *Cont.*

| Driver | Questions (Items) | Percentages | | | | | Mean | S.Deviation |
|---|---|---|---|---|---|---|---|---|
| | | Strongly Disagree | Disagree | Neutral | Agree | Strongly Agree | | |
| Security reasons | Q1- I moved to my current location seeking a safer environment. | 14.0 | 11.7 | 18.9 | 27.5 | 27.9 | 3.44 | 1.372 |
| | Q2- I moved to my current location because I was fleeing from a dangerous place. | 28.3 | 29.1 | 15.5 | 14.7 | 12.5 | 2.54 | 1.365 |
| | Q3- I moved to my current location because I felt threatened in my previous location. | 23.4 | 27.9 | 20.8 | 14.0 | 14.0 | 2.67 | 1.346 |
| Migration from inner-city | Q1- I moved from the inner city because suicide bomber attacks are concentrated there. | 10.9 | 24.2 | 26.0 | 18.1 | 20.8 | 3.14 | 1.296 |
| | Q2- I moved from the inner city because it is too crowded. | 12.1 | 25.7 | 25.3 | 21.9 | 15.1 | 3.02 | 1.252 |
| | Q3- I moved from the inner city because it's expansive. | 9.8 | 24.9 | 23.0 | 21.1 | 21.1 | 3.19 | 1.292 |
| Family nuclearisation | Q1- I moved to my current location because I started a family of my own. | 25.7 | 29.4 | 19.6 | 12.8 | 12.5 | 2.57 | 1.330 |
| | Q2- I moved to my current location because our family house was sold. | 28.3 | 29.1 | 20.8 | 11.7 | 10.2 | 2.46 | 1.291 |
| | Q3- I moved to my current location because I wanted to be independent of my parents. | 28.3 | 28.7 | 20.8 | 11.7 | 10.2 | 2.46 | 1.276 |
| Low commuting cost | Q1- I moved to my current location because public transportation is cheaper. | 26.8 | 25.3 | 24.5 | 12.8 | 10.6 | 2.55 | 1.296 |
| | Q2- I moved to my current location because the commuting cost is low due to my close workplace. | 21.1 | 27.2 | 21.9 | 15.8 | 14.0 | 2.74 | 1.332 |
| | Q3- I moved to my current location because gas prices are low, making commuting cheaper. | 22.3 | 27.2 | 24.5 | 13.6 | 12.5 | 2.67 | 1.301 |
| Development of transportation infrastructure | Q1- I moved to my current location because of the development of the transportation network. | 17.4 | 26.0 | 24.9 | 19.6 | 12.1 | 2.83 | 1.169 |
| | Q2- I moved to my current location because of the diversity in transportation modes. | 14.7 | 17.7 | 21.5 | 27.2 | 18.9 | 3.18 | 1.330 |
| | Q3- I moved to my current location because it has developed access to the inner city. | 15.1 | 21.5 | 26.0 | 21.1 | 16.2 | 3.02 | 1.298 |

Additionally, the data suggest that Q1 of security reasons and Q3 of low prices of land or dwelling have a high mean at 3.44 and 3.22. On other hand, the security reasons Q2, low commuting cost Q1, and family nuclearisation Q1, Q2 and Q3 means are low between 2.57 and 2.44, while the remaining drivers' questions mean is medium, between 2.61 to 3.20. Nevertheless, the highest standard deviation is for security reasons in Q1, Q2, and Q3 at 1.372, 1.365, and 1.346, respectively, while the lowest standard deviation is for rising income in Q1 at 1.208.

### 4.2.3. Urban Sprawl Impacts

The results shown in Table 6 show that several of the investigated impacts, such as less social interaction, agriculture land loss, and loss of natural habitat, have the highest total agree and strongly agree percentages of 47.5%, 46.6%, and 44.7%, respectively, followed by social segregation and insufficient health services at 42.3% and 41.2%. On the other hand, the study findings showed impacts with the highest total of disagreement and strong disagreement, such as mental health with a total of 61.7%, followed by insufficient educational facilities at 53%, insufficient public transport at 49.5%, less physical activity at 47.9% and pollution at 47.3%.

**Table 6.** Urban sprawl impacts descriptive results.

| Impacts | Questions (Items) | Percentage | | | | | Mean | S.Deviation |
|---|---|---|---|---|---|---|---|---|
| | | Strongly Dis | Disagree | Neutral | Agree | Strongly Agree | | |
| Social segregation | Q1- I find that people live segregated according to their tribe in my current location. | 15.8 | 18.5 | 16.6 | 29.4 | 19.6 | 3.18 | 1.368 |
| | Q2- I see people living segregated according to their class in my current location | 12.5 | 20.4 | 14.7 | 34.7 | 17.7 | 3.25 | 1.305 |
| | Q3- I find gated communities in my current location. | 29.4 | 23.8 | 12.1 | 18.9 | 6.8 | 2.41 | 1.273 |
| Less social interaction. | Q1-I find myself not frequently interacting with the members of my community. | 10.2 | 18.9 | 21.9 | 29.8 | 19.2 | 3.29 | 1.259 |
| | Q2- I do not spend much time in my community social gatherings. | 9.1 | 21.1 | 20.0 | 30.6 | 19.2 | 3.30 | 1.251 |
| | Q3- I don't find myself participating in arranging social events for my community. | 9.1 | 21.1 | 26.0 | 23.8 | 20.0 | 3.25 | 1.248 |
| Higher household cost | Q1- My transportation cost increased as I moved to my current location. | 17.0 | 29.8 | 20.4 | 21.5 | 11.3 | 2.80 | 1.270 |
| | Q2- My groceries cost increased as I moved to my current location. | 15.5 | 32.1 | 21.9 | 19.2 | 11.3 | 2.79 | 1.243 |
| | Q3- My rent got higher as I moved to my current location. | 18.1 | 26.4 | 18.9 | 21.5 | 15.1 | 2.89 | 1.343 |
| Higher public services cost. | Q1- My electrical and water bills increased as I moved to my current location. | 13.6 | 24.5 | 19.2 | 22.3 | 20.4 | 3.11 | 1.349 |
| | Q2- I spent more on waste management as I moved to my current location. | 12.8 | 27.5 | 25.3 | 17.7 | 16.6 | 2.98 | 1.279 |
| | Q3- I spent more on education as I moved to my current location. | 15.5 | 28.3 | 19.6 | 20.8 | 15.8 | 2.93 | 1.321 |
| | Q4- I have spent more on health care as I moved to my current location. | 17.4 | 26.8 | 23.8 | 18.5 | 15.8 | 2.84 | 1.293 |
| Loss of natural habitats. | Q1- My current location's natural trees decrease as people inhabit more land. | 9.8 | 19.2 | 20.8 | 29.1 | 21.1 | 3.32 | 1.273 |
| | Q2- In my current location, animal biodiversity decreases as people inhabit more lands. | 10.2 | 20.4 | 19.2 | 27.5 | 22.6 | 3.32 | 1.302 |
| | Q3- I find natural water resources decreasing in my current location as people inhabit more lands. | 16.2 | 21.9 | 27.9 | 18.5 | 15.5 | 2.95 | 1.294 |
| Agriculture land loss | Q1- I find that agricultural lands around my current location are being consumed to be inhabited. | 14.3 | 18.1 | 26.4 | 23.8 | 17.4 | 3.12 | 1.296 |
| | Q2- Agriculture land use around my current location is being converted to gain profit. | 10.6 | 13.2 | 25.7 | 29.4 | 21.1 | 3.37 | 1.249 |
| | Q3- I find that agriculture activities decreased or vanished around my current location. | 11.3 | 14.3 | 26.0 | 28.3 | 20.0 | 3.31 | 1.260 |
| Insufficient public transport | Q1- I find fewer public transport options in my current location. | 18.1 | 28.7 | 18.5 | 20.4 | 14.3 | 2.84 | 1.331 |
| | Q2- I have trouble finding public transport in my current location. | 21.1 | 31.7 | 17.7 | 18.9 | 10.6 | 2.66 | 1.290 |
| | Q3- The waiting period for public transportation in my current location is higher. | 18.9 | 30.2 | 14.7 | 22.3 | 14.0 | 2.82 | 1.347 |

**Table 6.** *Cont.*

| Impacts | Questions (Items) | Percentage | | | | | Mean | S.Deviation |
|---------|-------------------|------------|------|------|------|------|------|-------------|
| | | Strongly Dis | Disagree | Neutral | Agree | Strongly Agree | | |
| Insufficient educational facilities | Q1-I don't find educational facilities close to my current location. | 37.7 | 29.8 | 14.0 | 12.8 | 5.7 | 2.19 | 1.229 |
| | Q2- The quality of the educational facilities is low. | 20.0 | 28.3 | 20.0 | 21.1 | 10.6 | 2.74 | 1.287 |
| | Q3- The capacity of the available educational facilities is low. | 17.4 | 26.0 | 21.1 | 20.8 | 14.7 | 2.89 | 1.322 |
| Insufficient health facilities | Q1-I don't find hospitals close to my current location. | 24.5 | 23.8 | 17.4 | 21.5 | 12.8 | 2.74 | 1.374 |
| | Q2- The quality of the health facilities is low. | 17.0 | 17.4 | 22.6 | 25.3 | 17.7 | 3.09 | 1.346 |
| | Q3- The capacity of the available health facilities is low. | 14.7 | 20.0 | 18.9 | 29.1 | 17.4 | 3.14 | 1.327 |
| Unsafe environment | Q1- I find my current location to have a high crime rate. | 14.7 | 18.5 | 29.1 | 23.0 | 14.7 | 3.05 | 1.263 |
| | Q2- My current location does not have police patrols. | 13.6 | 23.4 | 20.4 | 23.0 | 19.6 | 3.12 | 1.336 |
| | Q3- I don't move around my home after dark. | 18.5 | 24.2 | 21.9 | 21.5 | 14.0 | 2.88 | 1.322 |
| Less physical activity | Q1- I walk less than I used to after moving to my current location. | 17.0 | 29.4 | 23.0 | 16.6 | 14.0 | 2.81 | 1.292 |
| | Q2- I don't usually walk to get my groceries. | 18.1 | 31.7 | 20.0 | 19.2 | 10.9 | 2.73 | 1.267 |
| | Q3- I find myself taking transport to almost every destination. | 20.8 | 26.8 | 20.4 | 18.9 | 13.2 | 2.77 | 1.330 |
| Mental health issues | Q1-I find myself more depressed as I move to my current location. | 33.2 | 29.4 | 17.7 | 11.3 | 8.3 | 2.32 | 1.270 |
| | Q2-I find myself more Anxious as I move to my current location. | 32.1 | 30.2 | 17.0 | 15.5 | 5.3 | 2.32 | 1.221 |
| | Q3- I often feel lonely since moving to my current location. | 31.7 | 28.7 | 19.2 | 14.3 | 6.0 | 2.34 | 1.231 |
| Pollution | Q1- I find water sources in my current location to be contaminated. | 22.6 | 24.5 | 21.5 | 20.0 | 11.3 | 2.73 | 1.318 |
| | Q2- The air quality is unhealthy in my current location. | 24.5 | 29.1 | 19.2 | 15.1 | 12.1 | 2.61 | 1.327 |
| | Q3- I find streets or places that become garbage disposal areas. | 17.4 | 23.8 | 25.3 | 18.9 | 14.7 | 2.90 | 1.306 |

Additionally, the data show that Q2 of agriculture land loss and loss of natural habitat of Q1 and Q2 have a high mean at 3.37, 3.32, and 3.32, respectively. This was followed by less social interaction in Q2, Q1, and Q3, which means between 3.25 and 3.30. On the other hand, the insufficient educational facilities mean is the lowest at 2.19, and also the social segregation Q3 mean is low at 2.41. The questions about the remaining impacts mean they are medium, between 2.61 to 3.20. Nevertheless, the highest standard deviation is for insufficient health facilities in Q1 at 1.374, followed by social segregation Q1 mean at 1.368, while the lowest standard deviation is for mental health in Q2 at 1.220.

### 4.3. Reliability and Validity

The reliability and validity of the questionnaire were tested by performing Cronbach's coefficients test for each driver and impact using SPSS 26, and the results are shown in Tables 7 and 8. The research also conducted the KMO and Bartlett's sphericity test on each measure, and the results are shown in Tables 9 and 10. The results of Cronbach's coefficients for the subscales are all greater than 0.70 in urban sprawl driver variables. In contrast, the impact variables, social segregation, and high household cost variables show a value of less than 0.7; thus, they were eliminated.

**Table 7.** Reliability analysis of each urban sprawl driver.

| Variable | Number of Items | Cronbach's Value | Cronbach's Value of Scale |
|---|---|---|---|
| Low prices of land or dwelling. | 3 | 0.794 | |
| Rising income | 3 | 0.818 | |
| Security reasons | 3 | 0.720 | |
| Migration from inner-city | 3 | 0.791 | 0.890 |
| Family nuclearisation | 3 | 0.745 | |
| Low commuting cost. | 3 | 0.863 | |
| Development of transportation infrastructure. | 3 | 0.833 | |

**Table 8.** Reliability analysis of each urban sprawl impact.

| Variable | Number of Items | Cronbach's Value | Cronbach's Value of Scale |
|---|---|---|---|
| Social segregation | 3 | 0.599 | |
| Less Social interaction. | 3 | 0.853 | |
| Household cost | 3 | 0.663 | |
| Higher public services cost | 4 | 0.849 | |
| Loss of natural habitats | 3 | 0.767 | |
| Agriculture land loss | 3 | 0.881 | |
| Insufficient public transport | 3 | 0.873 | 0.926 |
| Insufficient educational facilities | 3 | 0.776 | |
| Insufficient health facilities | 3 | 0.826 | |
| Unsafe environment | 3 | 0.776 | |
| less physical activity | 3 | 0.783 | |
| Mental health issues | 3 | 0.929 | |
| Pollution | 3 | 0.809 | |

**Table 9.** Urban sprawl drivers KMO and Bartlett sphericity test results.

| Variable | KMO | $X^2$ | df | Sig |
|---|---|---|---|---|
| Low prices of land or dwelling | 0.605 | 385.97 | 3 | 0.000 |
| Rising income | 0.711 | 278.94 | 3 | 0.000 |
| Security reasons | 0.601 | 215.22 | 3 | 0.000 |
| Migration from inner-city | 0.688 | 245.691 | 3 | 0.000 |
| Family nuclearisation | 0.649 | 193.769 | 3 | 0.000 |
| Low commuting cost | 0.715 | 385.71 | 3 | 0.000 |
| Development of transportation infrastructure. | 0.699 | 317.22 | 3 | 0.000 |

**Table 10.** Urban sprawl impacts KMO and Bartlett sphericity test results.

| Variable | KMO | $X^2$ | df | Sig |
|---|---|---|---|---|
| Social segregation | 0.528 | 153.30 | 3 | 0.000 |
| Less Social interaction | 0.705 | 363.05 | 3 | 0.000 |
| Household cost | 0.600 | 132.028 | 3 | 0.000 |
| Higher public services cost | 0.811 | 439.10 | 6 | 0.000 |
| Loss of natural habitats | 0.693 | 203.32 | 3 | 0.000 |
| Agriculture land loss | 0.740 | 427.76 | 3 | 0.000 |
| Insufficient public transport | 0.740 | 397.40 | 3 | 0.000 |
| Insufficient educational facilities | 0.648 | 241.61 | 3 | 0.000 |
| Insufficient health facilities | 0.718 | 292.133 | 3 | 0.000 |
| Unsafe environment | 0.688 | 221.02 | 3 | 0.000 |
| Less physical activity | 0.702 | 222.40 | 3 | 0.000 |
| Mental health issues | 0.762 | 632.94 | 3 | 0.000 |
| Pollution | 0.703 | 265.73 | 3 | 0.000 |

*4.4. Factor Analysis*

The study used principal component analysis and varimax rotation to conduct an exploratory factor analysis. The threshold for minimal factor loading was set at 0.50. Additionally, the commonalities of the scale, which measures the amount of variation in each dimension, were evaluated to guarantee sufficient levels of explanatory power. Exploratory factor analysis was utilised to minimise the number of factors and eliminate the measuring question with low loading. All commonalities less than 0.500 were excluded. Thus, the remaining factors and their loading are presented in Table 11.

**Table 11.** Exploratory factor analysis results of urban sprawl drivers and impacts.

| Urban Sprawl Drivers | | | Urban Sprawl Impacts | | |
|---|---|---|---|---|---|
| **Variable** | **Index** | **Factor Loading** | **Variable** | **Index** | **Factor Loading** |
| Low prices of land or dwelling | LP1 LP2 LP3 | 0.912 0.929 0.617 | Less social interaction | LSI1 LSI2 LS3 | 0.847 0.860 0.802 |
| Rising income | RI1 RI2 RI3 | 0.812 0.851 0.748 | Higher public services cost | HPSC1 HPSC2 HPSC3 HPSC4 | 0.788 0.802 0.786 0.740 |
| Security reasons | SR2 SR3 | 0.854 0.889 | Agriculture land and natural habitats loss | AGL1 AGL2 AGL3 NHL1 NHL2 NHL3 | 0.809 0.803 0.797 0.626 0.769 0.752 |
| Low commuting cost | LCC1 LCC2 LCC3 | 0.555 0.613 0.633 | Insufficient public transport | IPT1 IPT2 IPT3 | 0.838 0.848 0.782 |
| Development of transportation infrastructure | DTI1 DTI2 DTI3 | 0.740 0.844 0.837 | Insufficient health educational facilities | IHF1 IHF2 IHF3 IEF2 IEF3 | 0.727 0.749 0.805 0.610 0.694 |
| | | | Unsafe environment | USE1 USE2 USE3 | 0.735 0.745 0.601 |
| | | | Less physical activity | LPA1 LPA2 LPA3 | 0.591 0.768 0.680 |
| | | | Mental health issues | MH1 MH2 MH3 | 0.803 0.812 0.801 |
| | | | Pollution | POL1 POL2 POL3 | 0.746 0.741 0.641 |

4.4.1. Urban Sprawl Drivers

Confirmatory Factor Analysis

The study used a confirmatory factor analysis (CFA) on the five exploratory analyses of the selected drivers to create an urban sprawl driver measurement model. The findings are shown in Figure 6. The model levels were found to conform with the rule of goodness-of-fit, stating that the GFI, AGFI, and CFI levels are above 0.900, and CIMIN/DF is 1.783, which

is less than three; also, RMSEA is 0.054, which is acceptable, SRMR is 0.348 and PCLOSE is 0.306; hence, the model is determined to be fit [52].

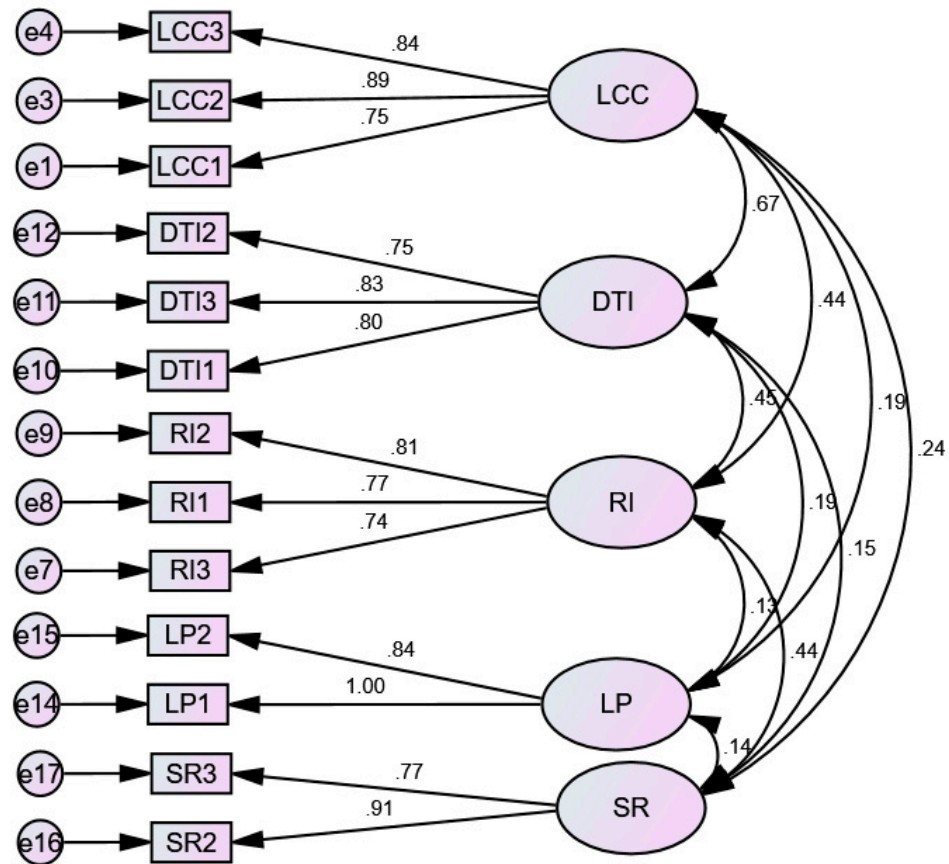

**Figure 6.** Proposal measurement model for urban sprawl drivers.

4.4.2. Urban Sprawl Impacts
Confirmatory Factor Analysis

The study used a confirmatory factor analysis (CFA) on the nine exploratory analyses of the selected impacts to create an urban sprawl driver measurement model. The findings are shown in Figure 7. The model levels were found to conform with the rule of goodness-of-fit, stating that GFI and AGFI are above 80,0, which is within the marginal limit, CFI levels are above 0.900, and CIMIN/DF is 1.633, which is less than three; also, RMSEA is 0.049, SRMR is 0.0525 and PCLOSE is 0.593; hence, the model is determined to be fit [52].

*4.5. Structural Equation Model for Urban Sprawl Drivers and Impacts in Mogadishu*
4.5.1. Mogadishu Urban Sprawl Drivers' Model

The Mogadishu urban sprawl drivers' model (Figure 8) is designed to assess the relationship between less commuting cost (LCC), development of transportation infrastructure (DTI), rising income (RI), low price of land and dwelling (LP) and security reasons (SR) and the main construct of urban sprawl drivers in Mogadishu. The models GFI and AGFI are above 900, which is good, and the CFI levels are 0.975, and CIMIN/DF is 1.722, which is less than three; also, RMSEA is 0.052, SRMR is 0.0590, and PCLOSE is 0.393; hence, the model fitness criteria findings indicated that the model achieved the requirement of a good fit [52]. Additionally, the hypothesised urban sprawl drives of Mogadishu, Somalia, were all proven to be significant (Table 12).

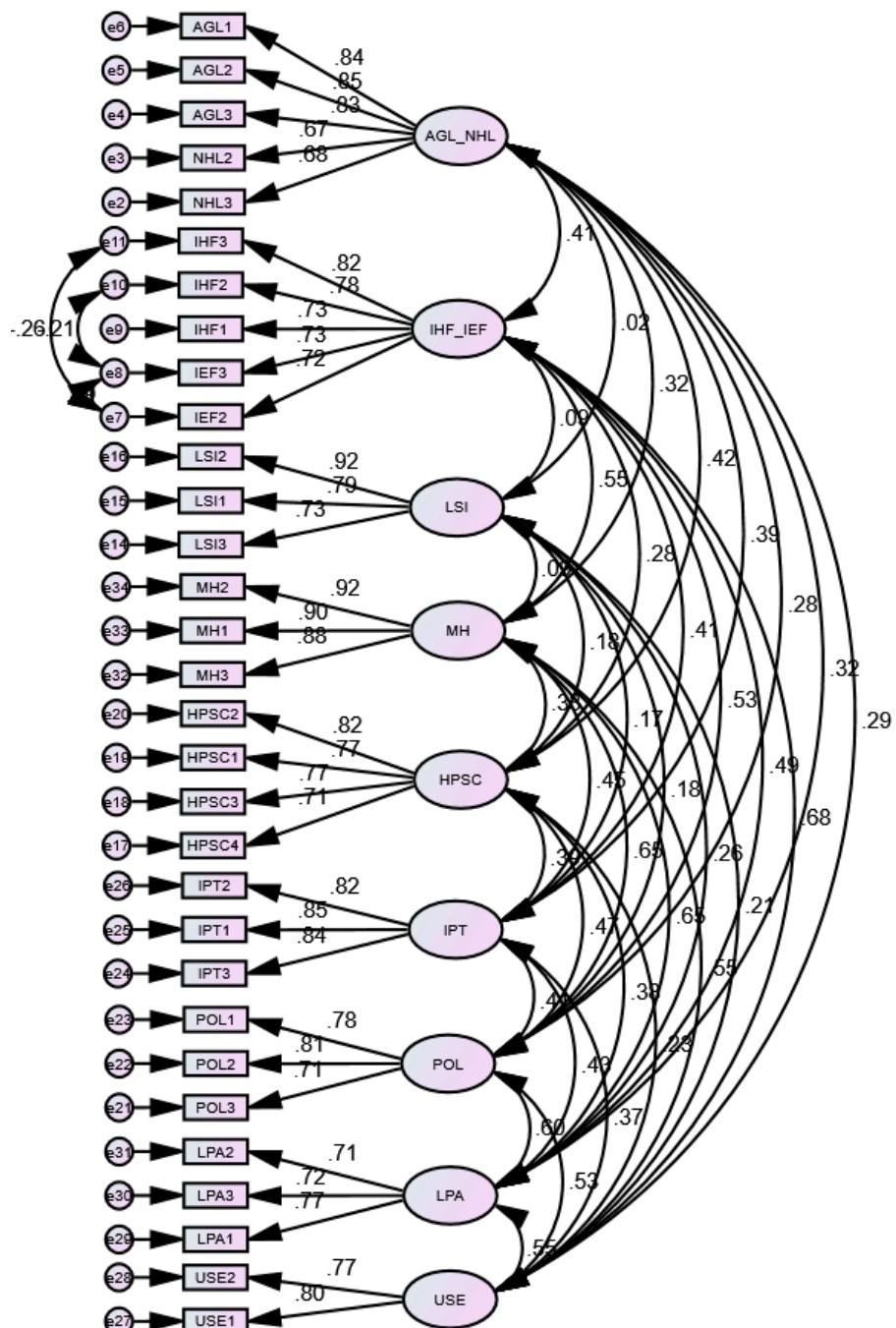

**Figure 7.** Proposal measurement model for urban sprawl impacts.

**Table 12.** Significance of the hypothesised urban sprawl drivers.

|  | Estimate | S.E. | C.R. | *p* | Result |
|---|---|---|---|---|---|
| LCC <— DRIVERS | 0.788 | 0.082 | 9.587 | *** | Supported |
| DTI <— DRIVERS | 0.811 | 0.083 | 9.783 | *** | Supported |
| RI <— DRIVERS | 0.547 | 0.076 | 7.203 | *** | Supported |
| LP <— DRIVERS | 0.299 | 0.088 | 3.408 | *** | Supported |
| SR <— DRIVERS | 0.373 | 0.095 | 3.937 | *** | Supported |

*** means $p \leq 0.001$.

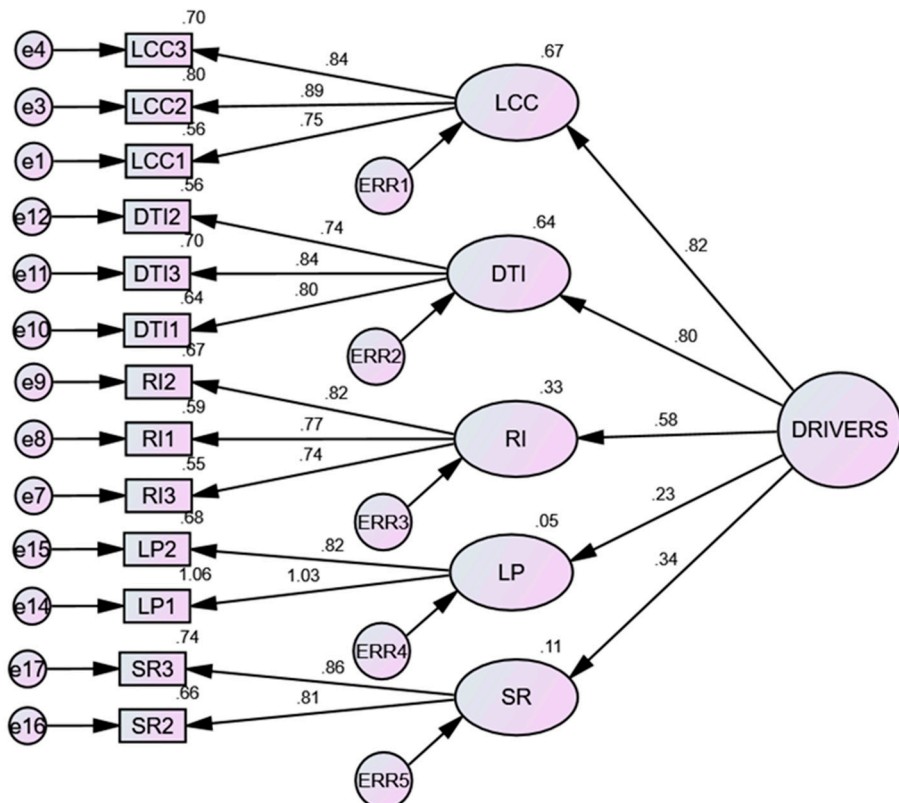

**Figure 8.** SEM for urban sprawl drivers in Mogadishu.

### 4.5.2. Mogadishu Urban Sprawl Impacts Model

The Mogadishu urban sprawl impacts model (Figure 9) is designed to assess the relationship between the main construct of urban sprawl impacts with agriculture land and natural habitat loss (AGL_NHL), insufficient health and educational services (IHF_IEF), less social interaction (LSI), mental health issues (MH), high public services cost (HPSC), insufficient public transport (IPT), pollution (POL), less physical activity (LPA) and unsafe environment (USE). GFI and AGFI are above 800, which is within the marginal limit, CFI levels are 0.935, and CIMIN/DF is 1.712, which is less than three; also, RMSEA is 0.052, SRMR is 0.067, and PCLOSE is 0.304; hence, the model fitness criteria findings indicated that the model achieved the requirement of a good fit [52]. Thus, it is a satisfactory fit. Additionally, the hypothesised urban sprawl impacts of Mogadishu, Somalia, were all proven to be significant (Table 13).

**Table 13.** Significance of the hypothesised urban sprawl impacts.

|  | Estimate | S.E. | C.R. | *p* | Result |
|---|---|---|---|---|---|
| AGL_NHL <— IMPACTS | 0.409 | 0.065 | 6.320 | *** | Supported |
| IHF_IEF <— IMPACTS | 0.690 | 0.072 | 9.638 | *** | Supported |
| MH <— IMPACTS | 0.848 | 0.068 | 12.449 | *** | Supported |
| LSI <— IMPACTS | 0.196 | 0.065 | 3.023 | 0.003 | Supported |
| HPSC <— IMPACTS | 0.462 | 0.069 | 6.692 | *** | Supported |
| IPT <— IMPACTS | 0.653 | 0.078 | 8.393 | *** | Supported |
| POL <— IMPACTS | 0.725 | 0.074 | 9.794 | *** | Supported |
| LPA <— IMPACTS | 0.756 | 0.075 | 10.074 | *** | Supported |
| USE <— IMPACTS | 0.721 | 0.077 | 9.387 | *** | Supported |

*** means $p \le 0.001$.

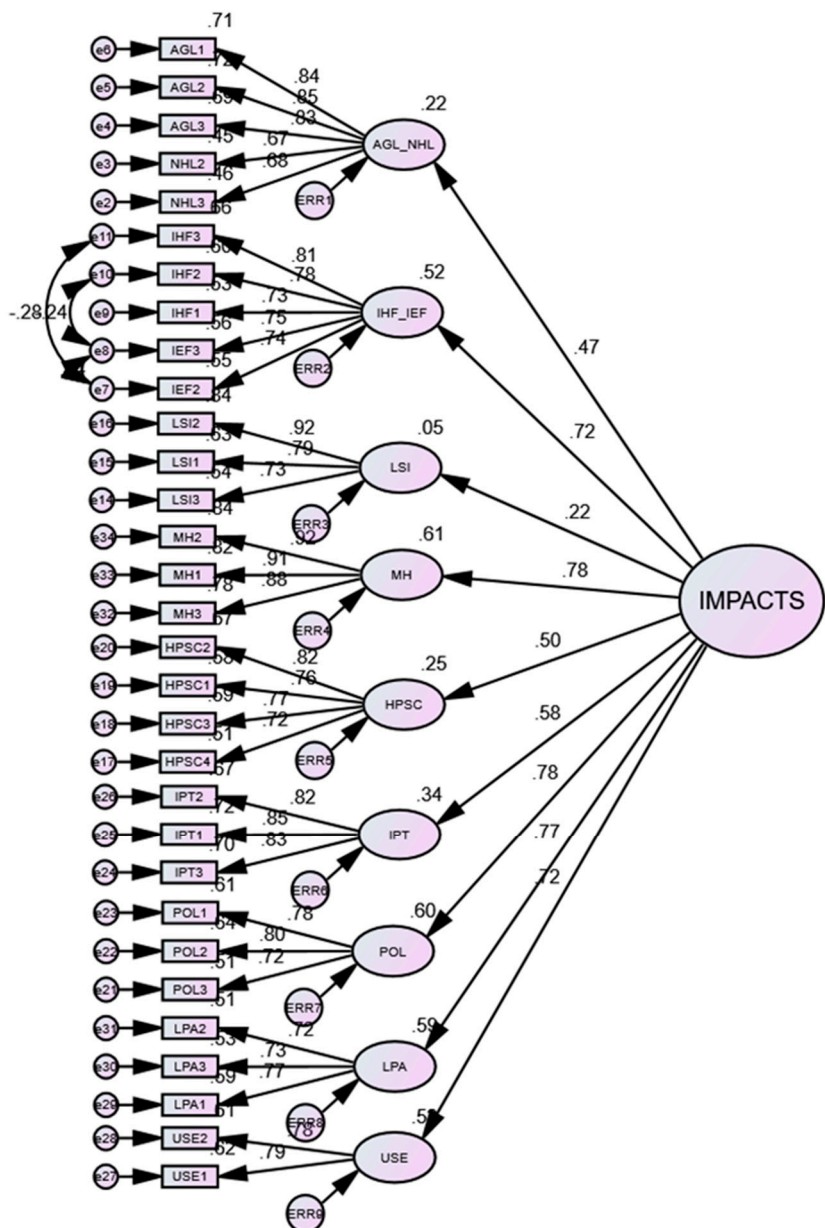

**Figure 9.** SEM for urban sprawl impacts in Mogadishu.

## 5. Discussion

The findings of this study showed that nine districts and three settlements had shown a sprawl pattern, mainly leapfrog and scattered. Additionally, the findings of both urban sprawl drivers in the Mogadishu models showed that all hypothesised drivers and impacts were significant. Meanwhile, when ranking the found drivers, the low price of land and dwellings (LP) is the primary driver of urban sprawl in Mogadishu, where 38.8% of participants confirmed it as their driver, followed by the development of transportation infrastructure (DTI) at 38.3%. Rising income was found to be 29.4%, security reasons (SR) at 27.6%, and low commuting cost (LCC) at 26.4%. On the other hand, when ranking the found impacts, less social interaction (LSI) is the major impact of sprawl in Mogadishu, where 47.5% of the participants stated it as an impact. Subsequently, 44.8% of participants stated that urban sprawl caused agriculture land and natural habitat loss (AGL_NHL), followed by an unsafe environment (USE) at 40.1% and insufficient health and educational services (IHF_IEF) at 38.2%. Moreover, high public services cost (HPSC) was at 36.9%,

insufficient public transport (IPT) at 33.5%, less physical activity (LPA) at 30.9%, pollution (POL) at 30.7%, and mental health issues (MH) at 20%.

Moreover, the findings of the land cover maps provide a visual representation of the expansion of urban areas in Mogadishu city, as well as the expansion pattern of built-up areas in 2006, 2013, and 2021. The findings of the produced 2006 map showed densely populated areas in several Mogadishu districts, including Hamarwayne, Hamarjajab, Hodan, and Hoolwadaag. Hamarwayne overpopulation is understood, since it initially settled in the 10th century and has been around for almost a millennium. Hamarjajab, Hodan, and Hoolwadaag, however, were not settled until later, as they were known as the historical core along with Hamarwayne [6]. The 2006 produced maps also show that the Karaan, Yaqshiid, Hawlwadaag, and Abdiaziz districts, as well as Shibis, had a high concentration of built-up areas. Grünewald [49] stated that those districts increased in the built-up area under Siyad Barre's rule from 1969 until 1991 when the city grew beyond its historical boundaries. After 2006, Mogadishu became the site of conflict between the transitional government and Al-Shabaab (AS) again [17].

The results of the 2013 map show that Wadajir expanded the most, followed by Hilwaa. There was a massive expansion north of the Kahda district, which was inhabited by people who fled to Mogadishu from other regions during the 2011 and 2012 Somali famine, when food donations from international aid organisations were distributed in Mogadishu [53]. Furthermore, 2021 produced maps showed that Kaaran, Wadajir, and Dherkeenly expanded rapidly and created multiple settlements such as Daarusalaam, Garisbaaley, and Gudadley, where land prices are much lower. This expansion happened after Somalia shifted from the transitional administration of Somalia to the Federal Republic of Somalia in 2012. At that time, Mogadishu appeared peaceful and quiet, as demonstrated by booming trade and the homecoming of Somalis from the diaspora [47].

Furthermore, the expansion in Mogadishu is seen to have different sprawling developing patterns such as leapfrog, scattered, and ribbon. The main urban sprawl patterns noticed are leapfrog and scattered, which were located in the northeast and north. Finally, urban sprawl is present in all Mogadishu districts and settlements except Hamarjajab, Hamrwayne, Shibis, Abdiaziz, Shangaani, Wartanabada, Waaberi, Howlwadaag, and Boondheere. The rest demonstrate varying degrees of spread and are appropriate for investigating urban sprawl drivers and repercussions in Mogadishu.

The finding of the SEM confirmed that less commuting cost (LCC) is a driver for urban sprawl in Mogadishu, where participants noted that they moved to their current location because of low transportation costs or low gas prices or closeness to the workplace, which is similar to the sprawl study findings in Iran [35]. In addition, similar to the findings of Karakayaci [26], this study also found that rising income (RI) is a driving factor for urban sprawl in Mogadishu, as a number of participants indicated that they moved to their current location after the increase in income to seek larger housing and better living conditions. Moreover, the finding of low prices of land and dwelling (LP) where participants stated they moved to their current location because they could not afford to live in the inner city and wanted a better and bigger dwelling they can afford, which is similar to what was seen in Egypt and Iran [35], and Moniya city in Nigeria [43]. Additionally, the development of transportation infrastructure (DTI) was found in Mogadishu as participants stated that they moved to their current location after a transportation network with access to the inner city was developed, similar to Iran's findings [35]. However, some drivers were not mentioned in the literature, such as security reasons (SR), which were investigated because of the Mogadishu history of conflict, and the finding proved it is a driver of urban sprawl in Mogadishu as the participants mentioned moving from their past locations fleeing from a dangerous place where they felt threatened.

On the other hand, the findings of the SEM proved eight urban sprawl impacts in Mogadishu, such as agriculture land and natural habitat loss (AGL_NHL), where participants stated they noticed agriculture lands are being consumed to be inhabited and agriculture activities decreasing, which is similar to Mysore city [29], and participants also

stated natural trees and water resources are decreasing in neighbouring areas as well as animal biodiversity. Additionally, less social interaction (LSI) was found as participants stated they do not frequently interact with the members of the community and do not participate in social gatherings, which is similar to Konya-Turkey's findings [5]. The finding also proved mental health issues (MH) are one of the impacts of sprawl in Mogadishu, as participants stated feeling more depression, anxiety, and loneliness after moving to their current location, similar to Mairena del Aljarafe municipality findings [7]. Moreover, the finding proved that high public services costs (HPSC) and insufficient public transport (IPT) are impacts of sprawl in Mogadishu as participants noted that they pay more for public services and face problems finding transportation similar to Sao Paulo in Brazil [45]. Nevertheless, pollution (POL) was found to be one of the impacts where participants noted that water resources are contaminated, worse air quality, and noticed a lot of streets and places becoming garbage disposal sites around their current location, although it is not as severe as in the case of Cairo, Egypt [1]. Additionally, less physical activity (LPA) was one of the findings where participants stated that they walk less and use transport to almost every destination, which aligns with Mysore's findings [54] and ultimately can cause obesity and heart problems, etc. Additionally, the study found insufficient health and educational services (IHF_IEF) where participants noted that they do not find hospitals and educational facilities close to their living places or that their quality is not good, similar to the findings in Brazil [6]. However, the unsafe environment (USE), which was investigated because of the Mogadishu history of conflict and context; the finding proved they are impacts of urban sprawl in Mogadishu as participants stated their current location has a higher crime rate and lack of police patrols, which is understandable knowing that each district has one police station.

## 6. Recommendation and Conclusions

The first step in mitigating urban sprawl and controlling the urban expansion process is to design and implement national and regional spatial planning [34]. However, the first urban master planning practice was held in 1937, which is the only city planning project in the Somali capital that has ever been completed in its entirety [49]. Thus, the first solution should be designing and implementing national and regional spatial planning to mitigate urban sprawl and ensure sustainable growth. Moreover, Li and Li [55] highlighted the need for planned land use and comprehensive city master plans to be developed by the government and planners working together, which in the case of Somalia does not exist as there is no land use planning carried out. Thus, comprehensive land use planning is crucially needed to mitigate urban sprawl and its impacts.

Nevertheless, planning authorities may address some of the study's findings, such as lower land and dwelling prices, by enacting affordable housing regulations that provide residents with a less expensive way of life in the inner city. Similarly, people moving to the city periphery because of security reasons can also be reduced if the city security level keeps improving. That being said, all these measures can work in Mogadishu. However, sprawl control would fail without a strong planning agency with the means and legislative backing to pay compensations, negotiate, enforce the law, and defend lawsuits against violators [16]. Moreover, as the finding of this study proved the existence and the impacts of urban sprawl, the government and international organisations should promote societal values that recognise the costs of putting personal gain before the public good along with putting in place methods for civic education, such as accountability for public space utilisation, sense of connection to the location of living and employment as Hosseini and Hajiilou [34] suggested. Additionally, it is crucial to enlighten people on the importance of ecological sustainability in the local environment and bioregion.

Finally, it is clear that urban sprawl is detrimental to the environment and the people of Mogadishu, there are no mitigation measures, and there is a lack of master planning and land use planning. Nonetheless, as the capital and economic centre of Somalia, Mogadishu will continue to attract large numbers of diaspora and people from other regions or rural

areas as the security situation continues to improve. Moreover, land and housing prices will continue to rise in the city centre and city outskirts, leading to further leapfrog and scattered development, similar to the Garasbaleey settlement. If the government does not take immediate action to mitigate urban sprawl, it will continue to accelerate and cause even more damage.

Despite its useful findings, this study has some limitations that should be considered, such as the lack of governmental population censuses or household surveys and the lack of high-quality satellite data that would contribute to a better clarity and understanding of sprawl. Nevertheless, participation was low in some of the most sprawling areas, such as Garasabaleey Settlement and Kahda District, due to isolation, community secrecy, and security concerns. Additionally, this study covered a wide number of drivers and impacts due to the lack of previous research on sprawl. That being said, further research is needed on the drivers and impacts of individual districts and settlements, particularly the Garasebaleey settlement and the Kahda district. Additionally, to better understand urban sprawl in Mogadishu, additional and more thorough research is needed on each of the drivers and impacts of urban sprawl identified in this study and other drivers and impacts using a more accurate approach, namely multi-temporal remotely sensed data via a landscape index in quantifying and classifying urban expansion. Finally, further research is needed in other major cities in Somalia, as urban sprawl can also be observed in other major cities.

**Author Contributions:** Formal analysis, M.O.H.; methodology, M.O.H., G.H.T.L., W.W. and N.R.; writing—original draft preparation, M.O.H.; validation, M.O.H. and G.H.T.L.; writing—review and editing, G.H.T.L., S.M. and P.C.L. All authors have read and agreed to the published version of the manuscript.

**Funding:** This research received no external funding.

**Data Availability Statement:** Not applicable.

**Acknowledgments:** The authors would like to thank the reviewers for their valuable comments and suggestions.

**Conflicts of Interest:** The authors declare no conflict of interest.

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
