# Peer review of "Urban Sprawl Patterns, Drivers, and Impacts: The Case of Mogadishu, Somalia Using Geo-Spatial and SEM Analyses"

_land, doi:10.3390/land12040783_

Round 1

Reviewer 1 Report

This is an overall well led research and well written paper. However, an improvement would be to better integrate the two main parts of the study, namely: i) the analysis of the spatial patterns, and ii) the analysis of the drivers and impacts of the urban sprawl of Mogadishu. In the paper, the presentation of these two parts are kept well distinct in the methods, results and discussions. At least in the discussion, the reader could expect that the patterns observed over the last 15 years (e.g. how much spatial changes, and where ?) would be interpreted in the light of the drivers identified in the SEM analyses for the present / recent past situation. Similarly, knowing the present trends in the drivers, it can be discussed what sprawl patterns can be expected in the near future.

Something that needs to be corrected is that the way some references are made. Several times, scientific literature is referenced using the names of the main authors, e.g. Hosseini and Hajiilou (2019), whereas according to journal style, the references are made using numbers in square brackets.

Finally, please check for common words starting with capital letters in the middle of sentences, e.g. Finally, Despite... (p5, l206)

Author Response

Reviewer 1

Comments:

  • This is an overall well led research and well written paper. However, an improvement would be to better integrate the two main parts of the study, namely: i) the analysis of the spatial patterns, and ii) the analysis of the drivers and impacts of the urban sprawl of Mogadishu. In the paper, the presentation of these two parts are kept well distinct in the methods, results and discussions. At least in the discussion, the reader could expect that the patterns observed over the last 15 years (e.g. how much spatial changes, and where ?) would be interpreted in the light of the drivers identified in the SEM analyses for the present / recent past situation. Similarly, knowing the present trends in the drivers, it can be discussed what sprawl patterns can be expected in the near future.

Response:

  • Thank you very much for the comments and words of encouragement. We have now revised our discussion where we have clearly and summarily highlighted the sprawl spatial changes, patterns and its locations (altogether 9 districts and 3 settlement. Besides, we also discussed the aforementioned sprawl in a more integrated manner by associating the sprawl with the factors and consequences identified using the SEM analysis (see Line 461-550). The discussion is also further enriched by inserting potential sprawl patterns in the city in the near future (please see Line 578 page 27).

Comments:

  • Something that needs to be corrected is that the way some references are made. Several times, scientific literature is referenced using the names of the main authors, e.g. Hosseini and Hajiilou (2019), whereas according to journal style, the references are made using numbers in square brackets.

Response:

2) Thank you for highlighting the issues. We have corrected the referencing and citation styles accordingly.

Comment:

3) Finally, please check for common words starting with capital letters in the middle of sentences, e.g. Finally, Despite... (p5, l206)

Response:

  • Thank you for highlighting the issues. We have corrected the errors throughout the paper.

Reviewer 2 Report

Dear editors and authors,

Thank you for providing me the opportunity to review the paper entitled “Urban Sprawl Patterns, Drivers, and Impacts: The Case of Mogadishu, Somalia Using Geo-spatial and SEM Analyses”. Knowledge of urban sprawl in developing countries is lacking, so this study offers an important lens to unveil the patterns, drivers, and impacts of urban sprawl in Mogadishu city. The paper is well-designed and easy to follow. However, I have concerns regarding the visual presentation part of built-up area maps. Because they are displayed separately, it is somehow difficult to follow and indicate where changes happen. I suggest that the authors use spatial analysis tools (for example, overlapping multi-year built-up area maps together) and quantities assessment methods to identify the location, timing, and type of urban sprawl patterns. This will greatly facilitate our understanding of the urbanization process in sub-Saharan African countries. In addition, there are many minor grammar/spelling errors in the manuscript. The paper will be much improved if the authors address these issues carefully.

Abstract

Line 17. What does “SSC” short for? Please provide the full name of abbreviation where it first appears.

Line 25. What factors and effects specifically?

Line 27. What are the potential implications for the authority and people of Mogadishu city under this continuous sprawl?

1. Introduction

Line 55. “There” should be written as “there”

Line 60. Same question. What does “SSC” mean?

2. Literature review

Please standardize the use of capitalization and punctuation. Below are some examples:

Line 95. “technological innovations…”

Line 99. “the built-up…”

Line 100. “built-area change…”

Line 116. ”sprawl is primarily…”

Line 122. “increasing population…”

Line 128. “lack of control of…”

Line 141. “, urban sprawl also…”

Line 147. “unclean water and bad sanitation…”

Line 137. One referencing paper is suggested to be added here, which reveals the intense relationship between urban expansion and cropland loss in China.

Tu, Y., Chen, B., Yu, L., Xin, Q., Gong, P. and Xu, B., 2021. How does urban expansion interact with cropland loss? A comparison of 14 Chinese cities from 1980 to 2015. Landscape Ecology36, pp.243-263.

3. Study Area, Data, and Methods

Line 179. “during”

Line 220. How many samples are used for the maximum likelihood algorithm and what is the classification accuracy?

Table 1. “Less physical activity”

Line 252. What are the specific criteria of the study’s requirements?

Line 254. What time frame?

Line 264. EFA?

4. Results and Findings

Line 268. Should be “4. Results and Findings”. Please also check the number of other sections.

Line 290. Can you quantitively identify the specific location and area of leapfrog and scattered pattern rather than descriptive analysis? See Liu et al. (2010) on classifying urban expansion types using landscape ecology metrics.

Liu, X., Li, X., Chen, Y., Tan, Z., Li, S. and Ai, B., 2010. A new landscape index for quantifying urban expansion using multi-temporal remotely sensed data. Landscape ecology25, pp.671-682.

Table 3 can be moved to the appendix.

Author Response

Reviewer 2:

Comment

  • Thank you for providing me the opportunity to review the paper entitled “Urban Sprawl Patterns, Drivers, and Impacts: The Case of Mogadishu, Somalia Using Geo-spatial and SEM Analyses”. Knowledge of urban sprawl in developing countries is lacking, so this study offers an important lens to unveil the patterns, drivers, and impacts of urban sprawl in Mogadishu city. The paper is well-designed and easy to follow.

Response:

  • Thank you for the kind comments. Appreciate it!

Comments:

  • However, I have concerns regarding the visual presentation part of built-up area maps. Because they are displayed separately, it is somehow difficult to follow and indicate where changes happen. I suggest that the authors use spatial analysis tools (for example, overlapping multi-year built-up area maps together) and quantities assessment methods to identify the location, timing, and type of urban sprawl patterns. This will greatly facilitate our understanding of the urbanization process in sub-Saharan African countries.

Responses:

2) Thank you for the suggestion. We have provided another map in which it clearly indicates the spatiotemporal pattern of the sprawl including its sprawl locations, years, and patterns (see Figure 5 page 10). We hope this will make readers identify and understand better the sprawl in terms of locations, time (years) and patterns.

Comments:

  • In addition, there are many minor grammar/spelling errors in the manuscript. The paper will be much improved if the authors address these issues carefully.

Response:

3) Thank you for pointing out the grammatical issues. We have checked and revised the errors accordingly.

Comments:

  • Abstract: Line 17. What does “SSC” short for? Please provide the full name of abbreviation where it first appears. Line 25. What factors and effects specifically? Line 27. What are the potential implications for the authority and people of Mogadishu city under this continuous sprawl?

Responses:

  • Thank you for the questions and suggestions. SSC means Sub-saharan Countries, and the abstract has been revised as you suggested. (See the newly revised Abstract)

Comments:

  • Introduction: Line 55. “There” should be written as “there”. Line 60. Same question. What does “SSC” mean?

Responses:

5)  Thank you. The above issues have been addressed where we have rephrased the sentence and removed the word SSC.

Comments:

  • On the Literature review. Please standardize the use of capitalization and punctuation. Below are some examples: Line 95. “technologicalinnovations…” Line 99. “the built-up…” Line 100. “built-area change…” Line 116. ”sprawl is primarily…” Line 122. “increasing population…” Line 128. “lack of control of…” Line 141. “, urban sprawl also…” Line 147. “unclean water and bad sanitation…”

Responses:

6) Thank you for highlighting the above issues of capitalisation and punctuation. We have revised the entire texts of the paper.

Comments:

7) Line 137. One referencing paper is suggested to be added here, which reveals the intense relationship between urban expansion and cropland loss in China. Tu, Y., Chen, B., Yu, L., Xin, Q., Gong, P. and Xu, B., 2021. How does urban expansion interact with cropland loss? A comparison of 14 Chinese cities from 1980 to 2015. Landscape Ecology36, pp.243-263.

Responses:

  • Thank you for the suggestion. We have added the reference to support the statement (see Line 141 page 4).

Comments:

8) On Study Area, Data, and Methods. Line 179. “during”. Table 1. “Less physical activity”. Line 264. EFA?

Responses:

8) Thank you for highlighting the issues. The above issues of capitalisation of words have been addressed and the acronym EFA has been described as Exploratory Factor Analysis.

Comments:

9) Line 220. How many samples are used for the maximum likelihood algorithm and what is the classification accuracy?

Response:

9) Thank you for the questions. The samples were 40 for each class (see line 220 page 5) and the accuracy of the classification is mentioned in table 3 (see page 10)

Comments

10) Line 252. What are the specific criteria of the study’s requirements?

Responses:

10) Thanks for the question. The criteria were being a member of a household who lives in a sprawl area (see Line 245-246 page 7)

Comments

11) Line 254. What time frame?

Responses:

11) Thanks for the question. The time frame of the questionnaire survey was 22 days from November 26, 2022 to December 18, 2022 (see Line 252 page 7)

Comments:

12) On Results and Findings. Line 268. Should be “4. Results and Findings”. Please also check the number of other sections.

Responses:

12) Thank you for pointing out the issues. We have revisited the paper and amended the number of sections where necessary.

Comments

13) Line 290. Can you quantitively identify the specific location and area of leapfrog and scattered pattern rather than descriptive analysis? See Liu et al. (2010) on classifying urban expansion types using landscape ecology metrics.

Liu, X., Li, X., Chen, Y., Tan, Z., Li, S. and Ai, B., 2010. A new landscape index for quantifying urban expansion using multi-temporal remotely sensed data. Landscape ecology25, pp.671-682.

Responses:

13) Thank you for your suggestion.  The suggested method is worthwhile to be explored but the current method adopted in this study was suitable and good enough since we have kappa accuracy to demonstrate quite clearly the patterns and locations of sprawl. However, to  improve the accuracy of the map showing the sprawl, another method using landscape index can be used to substantiate the current findings. In any case we have include this as part of our limitation of the study (line 593-597).

Comments

14) Table 3 can be moved to the appendix.

Responses:

14) Thank you for the suggestion. However, we still think it is more appropriate for this table to be put in the main text as it shows the kappa accuracy of the sprawl pattern identified. This provides the validity of the sprawl identified within the city and therefore achieves the first objective of this paper.